# Three-Dimensional Model of the Moon with Semantic Information of Craters Based on Chang’e Data

**DOI:** 10.3390/s21030959

**Published:** 2021-02-01

**Authors:** Yunfan Lu, Yifan Hu, Jun Xiao, Lupeng Liu, Long Zhang, Ying Wang

**Affiliations:** School of Artificial Intelligence, University of Chinese Academy of Sciences, No. 19 Yuquan Road, Shijingshan District, Beijing 100049, China; luyunfan17@mails.ucas.ac.cn (Y.L.); huyifan181@mails.ucas.ac.cn (Y.H.); zhanglong16@mails.ucas.ac.cn (L.Z.); ywang@ucas.ac.cn (Y.W.)

**Keywords:** China’s Chang’e project, impact crater, auxiliary annotation method, dataset of craters, deep learning, object detection, three-dimensional (3D) model

## Abstract

China’s Chang’e lunar exploration project obtains digital orthophoto image (DOM) and digital elevation model (DEM) data covering the whole Moon, which are critical to lunar research. The DOM data have three resolutions (i.e., 7, 20 and 50 m), while the DEM has two resolutions (i.e., 20 and 50 m). Analysis and research on these image data effectively help humans to understand the Moon. In addition, impact craters are considered the most basic feature of the Moon’s surface. Statistics regarding the size and distribution of impact craters are essential for lunar geology. In existing works, however, the lunar surface has been reconstructed less accurately, and there is insufficient semantic information regarding the craters. In order to build a three-dimensional (3D) model of the Moon with crater information using Chang‘e data in the Chang‘e reference frame, we propose a four-step framework. First, software is implemented to annotate the lunar impact craters from Chang’e data by complying with our existing study on an auxiliary annotation method and open-source software LabelMe. Second, auxiliary annotation software is adopted to annotate six segments in the Chang’e data for an overall 25,250 impact crater targets. The existing but inaccurate craters are combined with our labeled data to generate a larger dataset of craters. This data set is analyzed and compared with the common detection data. Third, deep learning detection methods are employed to detect impact craters. To address the problem attributed to the resolution of Chang’e data being too high, a quadtree decomposition is conducted. Lastly, a geographic information system is used to map the DEM data to 3D space and annotate the semantic information of the impact craters. In brief, a 3D model of the Moon with crater information is implemented based on Chang’e data in the Chang‘e reference frame, which is of high significance.

## 1. Introduction

The exploration of the Moon, the development of lunar resources and the building of a lunar base have been extensively studied for their huge potential significance. In addition, the analysis of the topography of the Moon underpins the building of a lunar robot base in the future to achieve a manned Moon landing. On Earth, geologists have employed several methods (e.g., field exploration) to measure topography and landforms and investigate their distribution and evolution. By complying with physical laws, geologists can understand the structure and motion properties of topography and landforms at different scales, as well as predicting and analyzing subsequent changes. The exploration of topography and landforms is critical to human survival. Unlike research on Earth, the study of the Moon’s topography, material composition, gravity distribution and morphology are determined overall by using aerospace technology. The analysis of the scanned data of the Moon refers to the mainstream method of investigating the topography of the Moon.

China’s Chang’e project plans to use the payload of the detector to acquire various data, and it has established databases including the “lunar image map” and the “whole-lunar-coverage digital elevation map” (DEM) (Figure 1). Notably, the DEM is an elevation map, so it almost displays binary images. These images lay a solid foundation for modeling and understanding the Moon. Lunar research based on Chang’e data is currently a research hotspot.

Impact craters are a critical feature of the Moon’s surface. Their size, shape and distribution involve sufficient geographic information.
Geologists are capable of discovering the laws of the origin, evolution and movement of the Moon by studying impact craters. Accordingly, impact craters should be detected by using accurate Chang’e data. For this detection, Lu et al. [1] proposed an efficient method called “LabelMe” to annotate the impact craters based on Chang’e data, which facilitates feature-learning with a deep learning method. By optimizing the original LabelMe method and employing it as the annotation tool, in this study we first seek to detect impact craters based on Chang’e data efficiently with our new deep learning network.


A three-dimensional (3D) model has richer information than a two-dimensional (2D) picture, and impact craters objectively have a rich 3D structure. Restoring the 3D structure of the Moon’s surface and showing the specific location of impact craters can obtain more intuitive results.


Based on Chang‘e data in the Chang‘e reference frame, this study aims to build a 3D model of the Moon with the semantic information of impact craters. The Moon has an oblateness of 0.0012 and is substantially close to the shape of a regular sphere. In this work, to build a more intuitive and beautiful 3D model of the Moon, a standard sphere acts as an approximation to restore the height of each point, and a four-step strategy is formulated and verified experimentally. Finally, a lunar surface model with both 3D structure and semantic information is built.


## 2. Related Work

### 2.1. Chang’e CCD Data

The Chang’e 2 stereo camera CCD was adopted to capture images at a 100 km orbital height to produce 7 meter and 50 meter digital orthophoto data with 844 and 188 frames, respectively. The respective frame data (e.g., a data file tif, the identical name coordinate information file tfw and the same name projection file prj) include 4196.11 GB of CCD data.


A fragment of information is stored in tif, twf and prj files. tfw and prj refer to text files, and the projection and position information of the files can be acquired by string parsing. tif represents a matrix that can be generated after the binary file is parsed.


All file names are set by complying with CE2_GRAS_xxx_aaa_bbbb_yyMzzzF_v.ext. CE2 is the Chang’e-2 mission. GRAS represents a data production unit. xxx indicates the product data type, i.e., DEM or DOM. aaa represents the resolution (i.e., 7, 20 and 50 m). bbbb represents the map number, where the first digit is a character and the last three digits are a number (e.g., C001). yyMzzzF represents the central geographic location of the slicing data, yy denotes a two-digit latitude value, *M* expresses a north latitude (*N*) or south latitude (*S*), zzz refers to a three-digit longitude value and *F* is an east longitude (*E*) or west longitude (*W*). *V* denotes the product version. In addition, ext is the product format suffix.


Figure 2 presents a graticule of Chang’e 50 meter data, and other resolutions are similar. Accordingly, the latitude and longitude corresponding to any point in the picture can be determined based on the tif and prj files with OSGeo.


### 2.2. Robust Annotation Tool

LabelMe [2] refers to a robust annotation tool that is being used extensively. As open-source software, it can directly load binary images in various formats (e.g., jpg, png) and JSON data containing encoded image strings. Moreover, comprising a range of annotation forms, this software can be used to annotate images with polygons, rectangles, circles, lines and points.


### 2.3. Lunar Impact Crater Detection Algorithm

Two scenarios in this project require the use of detection algorithms. The first is the suggested network in the auxiliary annotation tool, and the other refers to the object detection network for the crater detection. Object detection acts as a fundamental topic in computer vision research. Its basic definition can be simply considered as using a minimum horizontal rectangle to include all the pixels within which the object is located. For improvement schemes (e.g., polygon detection), this study will not go into detail.
In the following, the conventional crater detection method based on digital image processing and the universal object detection method based on deep learning are briefly described. In practice, the deep learning detection method was adopted for the lunar crater dataset.


#### 2.3.1. Impact Detection Based on Digital Methods

Existing methods employ various high-precision data to detect craters using spatial analysis and digital image processing (e.g., shape-fitting-based algorithm, Hough-transform-based algorithm [3] and machine-learning-based algorithm [4,5]). Although the mentioned methods to a certain extent contribute to the crater detection problem, it remains a challenge to improve robustness and efficiency.


#### 2.3.2. Object Detection Algorithm Based on Deep Learning

Over the past few years, deep convolutional neural networks (DCNN) have achieved great success in object detection tasks. Object detection models should consider the diversity of the appearance and scale of the objects. The classic methods consist of the two-stage method represented by Faster-RCNN [6] and the one-stage method represented by YOLO [7] SSD [8,9].


Faster-RCNN achieves high-precision object detection based on feature sharing and region proposal network (RPN) methods (e.g., R-FCN [10], Cascade RCNN [11], and MS RCNN [12]) have recently reached new heights in general data sets.


Compared with the two-stage method, the one-stage method lacks the proposal process and directly returns to the bounding box on the feature map. It exhibits a great advantage in speed, and it is therefore commonly employed in devices with fewer computational resources (e.g., various embedded machines).


As existing network quantization methods are leaping forward (e.g., ONNX [13], TVM [14]), neural networks will be accelerated.


### 2.4. 3D Construction of the Moon Model

At the macro level, some researchers have also used QuickMap technology to build models of the lunar surface, as shown in Figure 3. The schematic diagram covers four parts, i.e., orthogonal far-point observation, orthogonal near-point observation, south pole observation and north pole observation. These four sets of observations are capable of covering all areas of the lunar surface. In addition, the QuickMap technology can make the picture zoom in, zoom out and rotate, among other features, which helps observers to observe the topography of the lunar surface globally. However, the model complies with 2D pictures. Though it uses different perspectives to provide similar 3D observations, all pixels in the details pertain to a plane. After the model is zoomed, only 2D points can be observed, and the positions between the points cannot be understood. Moreover, the height change of a certain pixel relative to other surrounding pixels is impossible to determine. For instance, when using this model to detect impact craters, the 3D information of the impact craters is impossible to determine (e.g., the depth of impact craters and other spatial information). In summary, the model has a macroscopic view and a multi-directional observation angle, so the Moon can be analyzed in a comprehensive and high-level manner, whereas the 3D information of the lunar surface points is lacking, which is insufficient for building a 3D model with semantic information of impact craters.

## 3. Methods

To build a 3D model of the Moon with crater semantic information in the Chang‘E reference frame, we propose a four-step framework. First, to solve the problem of data annotation, an auxiliary annotation tool is implemented based on our existing study on the auxiliary annotation method [1] and the open-source software, LabelMe. The auxiliary annotation method is elucidated in Section 3.1. Subsequently, our auxiliary annotation tool is adopted to annotate the Chang’e data to create a dataset of the lunar craters, which is detailed in Section 3.2. Third, a quadtree decomposition method is employed based on deep learning detection methods to detect impact craters (as elucidated in Section 3.3). Lastly, a 3D model with impact crater information is built, as introduced in Section 3.4.


### 3.1. Auxiliary Annotation Method and Analysis

The annotation tools represented by LabelMe [2] have been extensively used in object detection, semantic segmentation, instance segmentation and other visual understanding tasks of data annotation. The full annotation method refers to a method of annotating all object areas in an image from scratch. To be specific, a picture is simply entered and annotated with tools (e.g., LabelMe). However, the full annotation method is significantly redundant for large-scale data. Thus, the auxiliary annotation method [1] is used to accelerate subsequent large-scale crater annotation. A neural-network-based auxiliary annotation method is implemented. The average image annotation time is reduced from 14 s to 5 s. The auxiliary annotation of pictures can be considered the process of pre-annotating pictures with an automated method, and then manually fine-tuning. This method saves considerable time by eliminating repeated mechanical operations during the annotation. In practice, our auxiliary annotation process is illustrated in Figure 4. Since the semantic information on the surface of the Moon is simple, based on the manifold law of data distribution, the identical type of high-dimensional data in nature is suggested to be concentrated in the vicinity of a low-dimensional manifold [15]. The work of the auxiliary annotation for the detection task aims to yield a function f(X)=Y, where $X\in R^{W\cdot H\cdot 3}$ denotes an RGB picture and $y_{i}\in Y$ is a sequence. For $y_{i}\in R^{4}$, $y_{i}=\left(x,y,w,h\right)$ represents a rectangular box, i.e., the smallest rectangular box containing all pixels of the object.


A common method for fitting the function *f* is to use detection networks (e.g., R-CNN [11,16,17]). This method has achieved great success on the COCO dataset, whereas sufficient computing resources are required, and the annotation task is commonly performed by using the desktop computer. The method proposed here uses a lightweight network to get real-time feedback in the case of only using CPU.


On the whole, the proposed method adds the auxiliary annotation module based on the existing full annotation method and carries out a human-friendly design for the label organization. In this experiment, the efficiency of the auxiliary annotation method is found to be 2.8 times of that without auxiliary annotation.


#### 3.1.1. The Algorithm Processes

Set $I\in R^{W\times H\times 3}$ as a RGB image, and set manual annotation operation as *F*. So, the time of a manual annotation operation can be expressed as $T_{completely}=F\left(I\right)$. For the auxiliary process, the time can be expressed as $T_{proposal}=P\left(A\left(I\right)\right)$, while *P* represents the operation of the fine-tuning process for the proposed object box A(I). By auxiliary annotation, the label annotator only needs to fine-tune the proposed object box manually.


#### 3.1.2. Quantification

The purpose of quantification is to describe the composition of the annotation time. Figure 5 and Figure 6 show the process of annotating once from left to right. The bottom line of these two figures is a breakdown of the up line. For such a pipeline process, we use a quantitative method to measure time (the full annotation method is shown in Figure 5, auxiliary method is shown in Figure 6). For the full annotation method, it can be quantized into the above pipeline. The specific steps are elucidated below:A. Annotation software loads images;B. Annotator preliminarily observes the picture according to the experimental principle of human–computer interaction;C. Confirm the goals inside the images and start annotating;D. This process is annotating and classifying. Specifically, it can be divided into several E, F processes;E. Confirm the target boundary;F. Put a label on the confirmed boundary;G. Save and annotate the next one.

As indicated from the actual operation, there is no method for parallelizing pipeline stages in the conventional full annotation method. Moreover, the pipeline for the process with auxiliary annotation is illustrated in Figure 6. The detailed steps are as follows:
1. Recommendation generation of the proposal network;2. Fine-tuning of the suggested label;3. Target confirmation and annotating.

#### 3.1.3. Architecture of the Auxiliary Network.

According to Figure 7, the complete auxiliary network comprises Input, Proposal Network, Output and Visualization that draws the bounding-box on the input image. The auxiliary network takes the picture as the ordinary input, while obtaining the output via the Proposal Network. The output contains the bounding-box information of the target to be annotated. Lastly, the output result is visualized on the picture. The Proposal Network is a real-time object detection network with the only CPU. Among the whole process, the Proposal Network is recognized as the most time-consuming part.

The Proposal Network employs a similar framework as an SSD, while the backbone is replaced with MobileNets-v1 to enable it to run on a PC with the only CPU in real time. The SSD algorithm allocates the feature block in the feature map of each layer to a classifier and a regressor. Its major computational complexity is in the feature extraction part. This is because in this task, the mAP displays a non-linear relationship with the annotation time of the auxiliary process.

Based on experiments, the SSD detection algorithm with MobileNet as the skeleton can give real-time feedback suggestions, so MobileNet-SSD was adopted as a lightweight detection network, and then a complete auxiliary suggestion method was formed.


### 3.2. Chang’e Data Annotation and Quantitative Analysis

After studying the auxiliary annotation method, we implemented auxiliary annotation software based on MxNet [18] and LabelMe [2], and the results after processing are shown in Figure 8. We annotated some segments with auxiliary annotation methods, which were then to be used as the training set data. The characteristics of the statistical data themselves are crucial for detection. For instance, in some current pedestrian detection work, the length-to-width ratio of a person fluctuates within a small interval. Using the prior characteristics of the detection targets can enable design of a better detection algorithm [19].
Impact craters also have many a priori features, such as the fact that most of them are oval. At the same time, craters also have local distribution characteristics; for example, there are many small craters around a large crater.


In practice, we counted the number of impact craters in each standardized picture, and obtained the statistical distribution of the length and width. Then, we compared the distribution of the shapes of the objects from Chang’e with those in a common dataset, e.g., COCO [20] (for more details refer to Section 4.2). In summary, the distribution characteristics of impact craters in the Chang’e data were very important for the design of the algorithm.


### 3.3. Detection Algorithm of Impact Craters

Conventional crater detection methods rely heavily on manual tuning and can only perform well on specific data. In addition, these methods usually fail in the detection of large-scale data.


On the other hand, object detection algorithms based on deep learning methods exhibit high performance on the general dataset, whereas the performance is difficult to test on the lunar crater dataset due to the lack of crater data. However, migrating the common detection algorithms (e.g., [12,21,22,23]) will be subject to the following difficulties:
The resolution of the original picture is extremely high.The field of view is wider.There are more objects in each image.Impact craters of different sizes are larger.Impact craters are mostly elliptical and generally use rectangular representation.

Unlike COCO and other pictures in daily life, which are mostly around 1024×1024, CLEP data has an extremely high resolution, usually around $3*10^{4}\times 3*10^{4}$, so many algorithms designed for COCO are subject to problems regarding large images. It is a crucial issue to design an algorithm that can guarantee the accuracy of large images and are compatible with general image processing algorithms.


#### 3.3.1. High-Resolution = Low-Resolution + Tree Structure

Since the resolution of the Chang’e data is significantly high, the Chang’e data cannot be directly used as the input during training and inference. To ensure normal training, we used quadtree (illustrated in Figure 9) to decompose the Chang’e data. Assuming input_size=4096 and detection_size=1024, the height of the tree is height=log(input_size/detection_size)+1=3. The number of leaf nodes is $node_{leaf}=4^{height-1}=16$. The number of non-leaf nodes is $node_{parent}=\left(4^{height-1}-1\right)/3=5$.

#### 3.3.2. Converting Large Pictures into Quadtrees

In the framework here, the baseline method of high-resolution image decomposition uses a sliding window. Compared with the quadtree decomposition method proposed in this study, the sliding window method cannot effectively identify targets that span multiple regions. Assuming that the blue area in Figure 10 represents the target to be recognized, if the sliding window method is used, only a part of the target can be recognized at a time, and there is no global view. Accordingly, the quadtree decomposition method is more robust.

### 3.4. Construction of the 3D Model with Impact Crater Information Based on GIS Coordinate Mapping

The geographic information system (GIS) is a very important and special spatial information system. It is a technical system that stores, manages, calculates, analyzes and describes the spatial information of the entire planet (e.g., the Earth and the Moon) with the support of computer software and hardware. Chang’e raw data has a strict format, and we can use GIS software (e.g., ArcGIS [24] and openGIS [25]).

In this study, we used the ArcGIS software for visualization and the OSGeo package for coordinate mapping. In Figure 11, we show an example of the Chang’e-2 DOM-7m North Pole, in which each grid is divided into frames N01 to N36. We can use GIS to combine framing pictures, coordinate mapping, and other operations.

The current operation here acquires the crater semantic information in two dimensions (Figure 12). Before coordinate mapping, the prj and tfw files should be read. The prj file has a Key-Value format. For instance, the sentence [“Moon_2000_IAU_IAG”,1737400.0,0.0] specifies that the radius of the Moon is 1,737,400 m, and the oblateness is 0.0, so it is a standard sphere. GEOGCS[“GCS_Moon_2000”... expresses the geographic coordinate system of the Moon. The tfw file defines the relationship between image pixel coordinates and actual geographic coordinates. There are six values in the tfw file, i.e., pixel resolution in the x-direction, x-direction rotation factor, y-direction rotation factor, pixel resolution in the y-direction, x-coordinate of the pixel center in the upper left corner of the grid map and y-coordinate of the pixel center in the upper left corner of the grid map.

In summary, using osgen to parse the prj and tfw files can yield a one-to-one mapping function between the image and latitude and longitude.

## 4. Experiments

### 4.1. The Relationship between mAP and Auxiliary Cost

In this experiment, the relationship between the auxiliary net with different mAP and the cost time was explored. Figure 13 suggests that with the increase in the mAP, the auxiliary network is more accurate and takes less time to annotate. At the annotating stage, the sum time of all five annotators is averaged when each annotator annotates five sets of photos, i.e., complete set (label an image without any supplementary information), auxiliary set (input is the result of a proposal net and has some weak annotation) and observed set (only observing the image and without annotation). Table 1 lists the improvement of the proposal net for annotation of Chang’e data. The auxiliary network of the auxiliary method is SSD-MobileNet, which is 71.0 at mAP@0.5. The time here refers to the average time to operate 50 pictures.

### 4.2. Dataset Analysis

The dataset is the basis for the data-driven method. Comparing the characteristics of different datasets is critical for adjusting the detection task. In the present section, the dataset is evaluated in two ways. The existing crater dataset consists of two parts, one of which refers to a dataset that astronomers have labeled, and the other is our own. Moreover, the Robbins dataset published in [26] can be obtained and most of the impact craters in this part of the data have a radius larger than 3 km. As impacted by the high resolution of the Chang’e data, numerous small impact craters are not covered in this dataset. Figure 14 presents the set relationship between the two existing sets of craters. Large rectangles represent all craters. The respective small blue rectangles represent a crater in a frame. The green area represents the larger collection of craters, and the red area indicates the collection of craters we annotated. We refer to all the green areas as Robbins and the pure red areas as Chang’e.

The number of regions of interest in each image is inconsistent with each data set. We calculated statistics for COCO, Robbins and our own annotated data, by counting the number of target frames in each image and calculating the percentage of the images containing the corresponding number of targets in the total dataset (Figure 15). The size of the target object change is recognized as a problem in the detection. In the quantification of the detection bounding box size, the two impact crater data sets are compared with COCO (Figure 16).


### 4.3. Crater Detection

After analyzing the crater data set, we employed deep learning methods to train the model and evaluate it. First, the SSD framework was adopted to evaluate a dataset that contained only our own labels. The result of the evaluation is listed in Table 2. Next, Faster-CNN with higher accuracy on COCO was adopted for evaluation. The evaluation data are all presented by photos with labels. For our unlabeled photos, many small impact labels are missing. As revealed from the results, if some photos are missing labels, the overall detection result will be affected.

This study suggests that the mAP on COCO is not directly related to the mAP of the impact craters, demonstrating that a network that performs well on COCO may not perform well on the impact crater dataset. This result is probably explained by the fact that the impact crater data are relatively single, and the network is prone to overfitting.

A learning rate of 0.001 was used in the experiment and the batch size was set to 32. Each model was trained for 240 epochs. Using the SGD optimizer, at the 160th and 200th steps, the learning rate becomes 0.1 times the previous one. All experiments were performed on a machine with two TITAN V GPUs.

### 4.4. Create 3D Model

The coordinate mapping method was adopted to map 2D information to 3D. A fragmented 3D point cloud was formed, as shown in Figure 17.

Take CE2_GRAS_DEM_50m_K003_49S120W_A as an example. We show the 3D model with crater information in the form of a point cloud. It is worth noting that to display the point cloud normally, the original point cloud information is downsampled. Since the original point cloud is very large, there are more than 100 million points. Here, the point cloud is presented after 64 times downsampling, i.e., the original point cloud is selected only when the horizontal coordinate of the 2D axis is a multiple of 4096. Lastly, each segment is stitched on a standard sphere to build the model (Figure 18).


The distribution of impact craters exhibits the following characteristics. (1) The impact craters are extensively distributed on the surface of the Moon, and a big impact crater is usually surrounded by many small craters, which constructs a local feature of the distribution. (2) The sizes of impact craters are quite different. As shown in the figure, the largest impact crater size takes up nearly 1/4 of its width and length, while some small impact craters are only one-thousandth of the large impact craters in size. (3) The fluctuation of the impact craters is small compared with the scale of the framing. As shown in the second and third rows of the figure, viewing the point cloud from the side, the fluctuation is very narrow.

Figure 17 also shows the 3D model of the semantic information of impact craters in this frame. This model has many potential applications. For instance, (1) the distribution of impact craters can be judged by complying with the semantic information of impact craters; (2) it can be based on 3D structure statistics on the relationship between crater depth and radius; (3) more detailed classification of craters can be performed based on 3D structure (e.g., judging the age of craters). These potential applications are of high significance for lunar geology research.

Based on the framing process, this study merges the coordinates of each framing to form a complete sphere (Figure 18a). After merging all the point clouds of the frame, the number of point clouds has reached billions, and these cannot be loaded and visualized with a personal computer. Accordingly, in this study, downsampling was performed in the experiment, and the vertical and horizontal axes of the frame data were every 64 pixels. Each unit was sampled once, so the overall data volume became 1/4096 of the original, which could then be displayed with a personal computer. Figure 18b adopts an identical strategy for visualization, while annotating the large impact craters. The specific method is to select the crater with a larger radius in the crater detection result. The crater is represented as a four-dimensional vector composed of the center point and the long and short axes. During the mapping, the mapped point is traversed in the crater list. If the point belongs to a certain impact crater, the point is annotated to distinguish impact craters from non-impact craters. Notably, the method implemented in this study is capable of achieving coordinate mapping without downsampling, as well as ensuring that the resolution of the 3D spatial structure matches the resolution of the original Chang’e data. Here only downsampling was performed for the visualization task, and other research was not required.

## 5. Discussion

Data annotation is critical to the data-driven approach. In this study, the auxiliary annotation method was used to annotate the impact craters on the Moon. We also presented the relationship between mAP and annotated time. The auxiliary network adopted in this study was trained with full supervision. Subsequently, if there are more objects to be detected, the auxiliary method can refer to several semi-supervised and weakly supervised methods.

In the detection method for impact craters, this study employed quadtree decomposition to solve the problem of excessive resolution. Moreover, decomposition structures (e.g., nine-node-tree) can be adopted here, although it makes no essential difference. However, the key point is how to decompose high-resolution pictures.

Lastly, the 3D model here is represented by a point cloud, since the point cloud contains the most primitive information, and a mesh can be used in some visual displays as well. There are many algorithms capable of converting a point cloud into a mesh grid.

## 6. Conclusions

Starting from Chang’e data, we designed a series of practical and novel methods and built a three-dimensional (3D) model of the Moon with both 3D structure and crater semantic information. These methods comprised the auxiliary annotation method of the impact craters, the construction method of the Moon impact pit dataset, the detection method of the Moon impact craters and the mapping method of two-dimensional (2D) lunar coordinates to 3D coordinates. To be specific, the auxiliary method designs the auxiliary network to expedite the annotation of the data and determines the relationship between mAP and annotation time, which presents several ideas for subsequent researchers. When annotating the data set, we analyzed the distribution characteristics of impact craters and studied the distribution rules of impact craters. Accordingly, an important reference is provided for future researchers to study and analyze impact craters. In the method of detecting craters, this study has solved the problem of the resolution of Chang’e data being too high. The designed quadtree structure is capable of ensuring both the visual field of detection and the normal operation of the network. Lastly, coordinate mapping was employed to map 2D information to 3D, forming a lunar surface model that has both 3D and semantic information.

In brief, in this study we proposed a set of methods capable of using original Chang’e data to construct a 3D model of the Moon with crater information. This series of methods is of great significance for both geographical research experts and popular science enthusiasts.

## Figures and Tables

**Figure 1 sensors-21-00959-f001:**
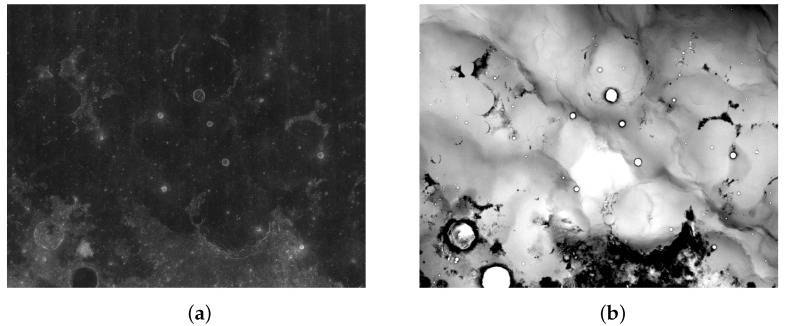
Examples of the two types of image of area $18^{{}^{\circ}}$ S∼$0^{{}^{\circ}}$ N $306^{{}^{\circ}}$ W∼$324^{{}^{\circ}}$ W: (**a**) digital orthophoto image (DOM) image; (**b**) digital elevation map (DEM) image.

**Figure 2 sensors-21-00959-f002:**
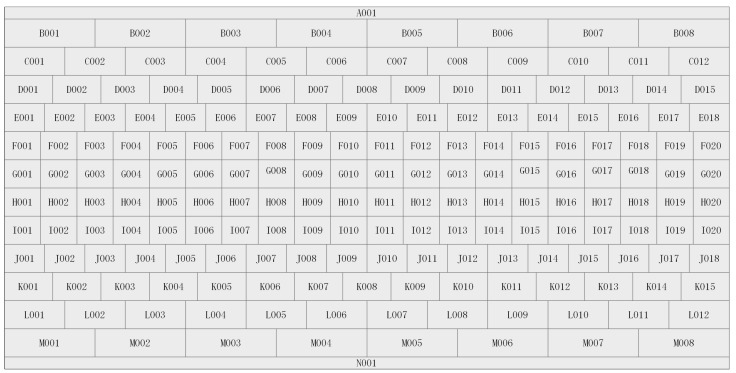
The complete Chang’e 2 CCD 50 m graticule.

**Figure 3 sensors-21-00959-f003:**
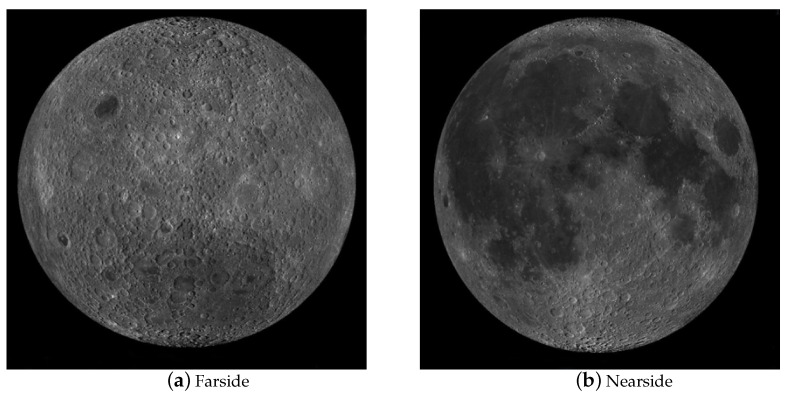
Schematic diagram of the QuickMap Moon model.

**Figure 4 sensors-21-00959-f004:**
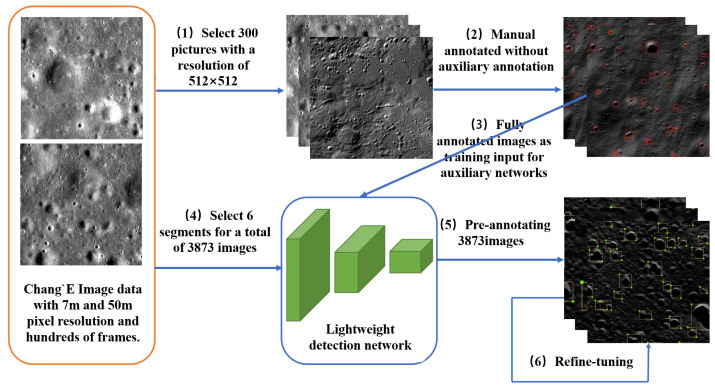
Illustration of the auxiliary annotation process. (1) The selection of 300 divided images with a resolution of 512×512; (2) the annotation process of the 300 pictures with the annotation tool; (3) the fine-tuning process of the pre-trained lightweight detection network using labeled images; (4) the selection (6 frames) and decomposition process (splitting the selected frames down into 3873 pictures with a resolution of 512×512); (5) the pre-annotating of the exploded pictures; (6) the manual fine-tuning process of pre-annotated pictures.

**Figure 5 sensors-21-00959-f005:**
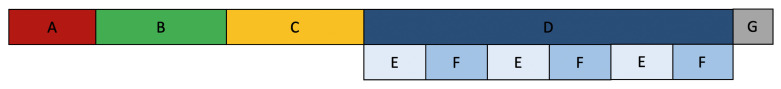
Full annotation method pipeline.

**Figure 6 sensors-21-00959-f006:**
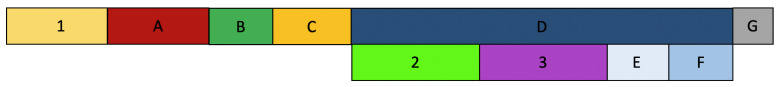
Auxiliary annotation method pipeline.

**Figure 7 sensors-21-00959-f007:**
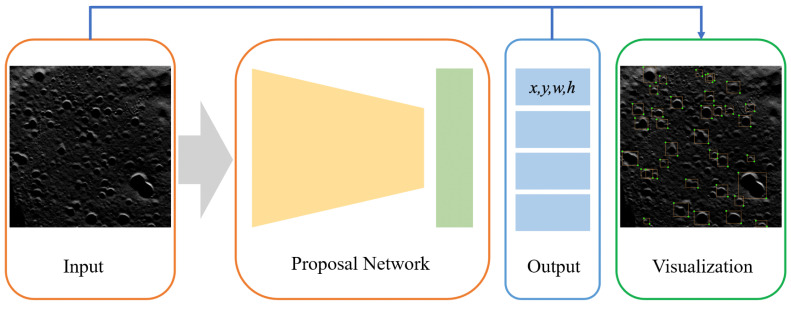
Structure of the auxiliary network.

**Figure 8 sensors-21-00959-f008:**
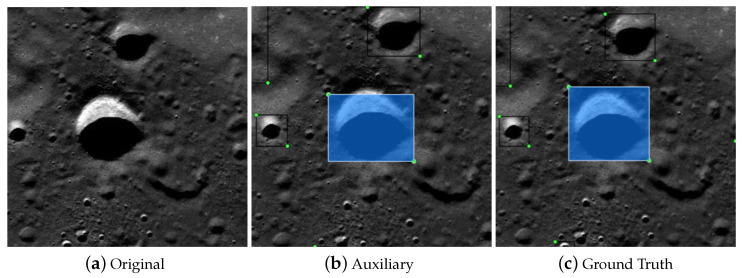
Visualization of a Moon image under different processing processes: (**a**) the input picture; (**b**) the pre-labeled picture using the auxiliary network; (**c**) the manually fine-tuned picture.

**Figure 9 sensors-21-00959-f009:**
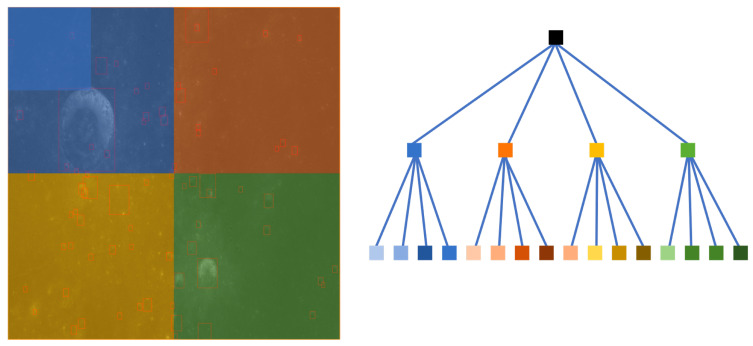
Tree decomposition of Chang’e data.

**Figure 10 sensors-21-00959-f010:**
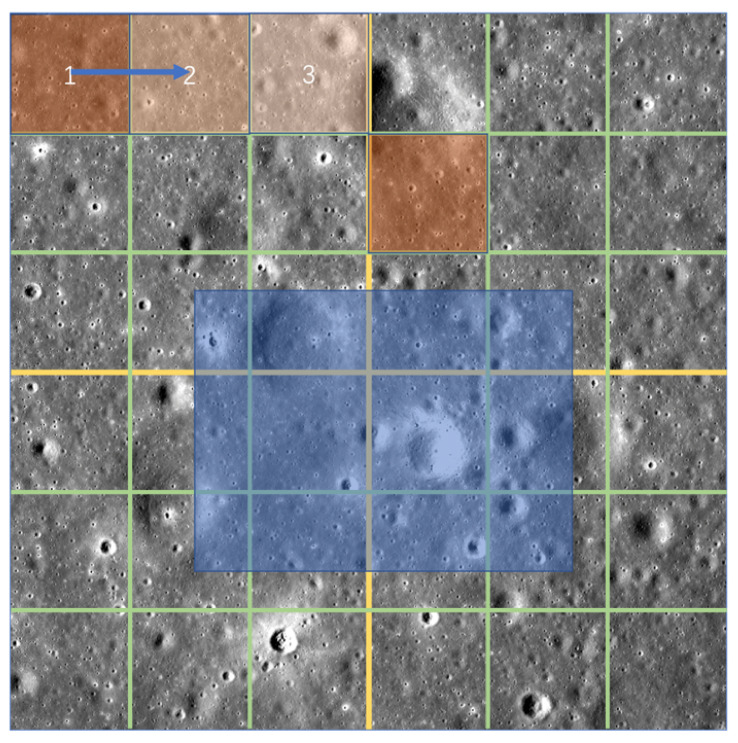
Decomposition diagram of impact crater detection.

**Figure 11 sensors-21-00959-f011:**
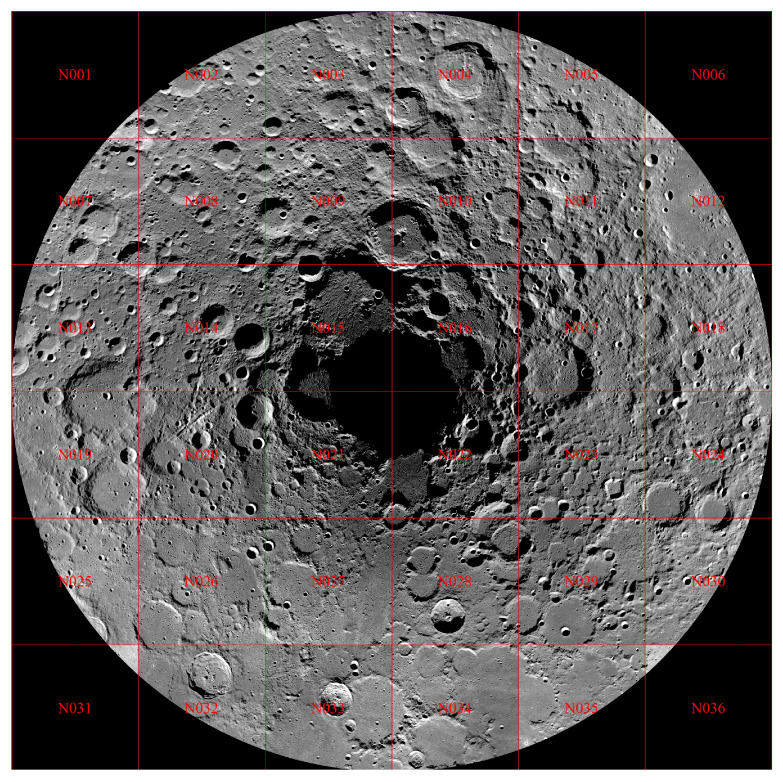
The arctic framing diagram of Chang’e DOM-7m.

**Figure 12 sensors-21-00959-f012:**
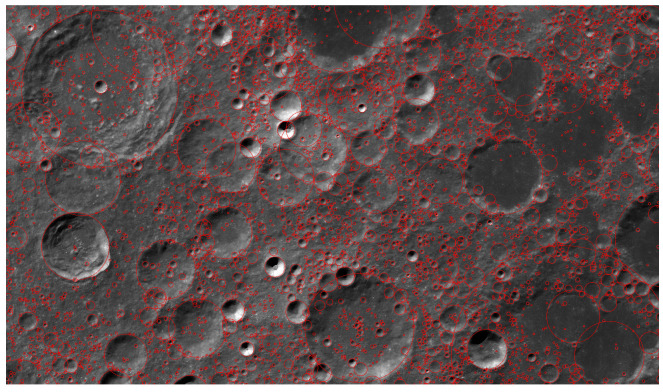
Image of lunar surface with crater coordinates.

**Figure 13 sensors-21-00959-f013:**
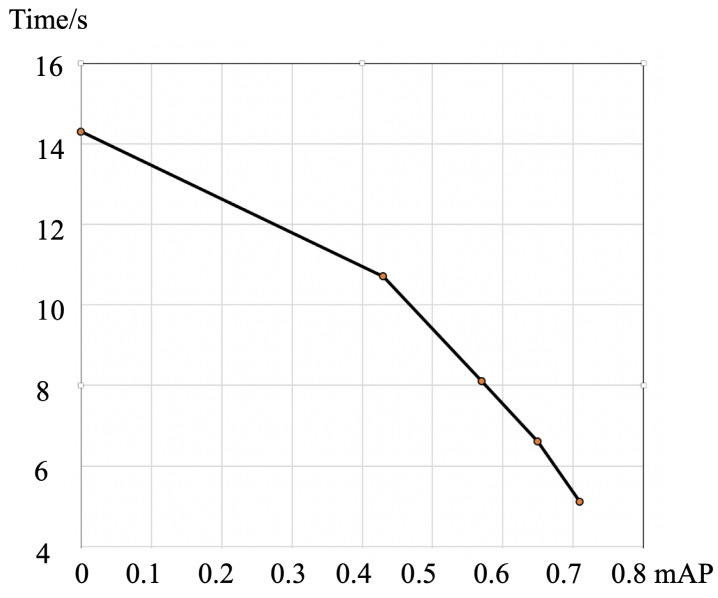
The relationship between different mAPs of the auxiliary net with cost time. The y-axis is the annotation time and the x-axis is the mAP of different proposal networks.

**Figure 14 sensors-21-00959-f014:**
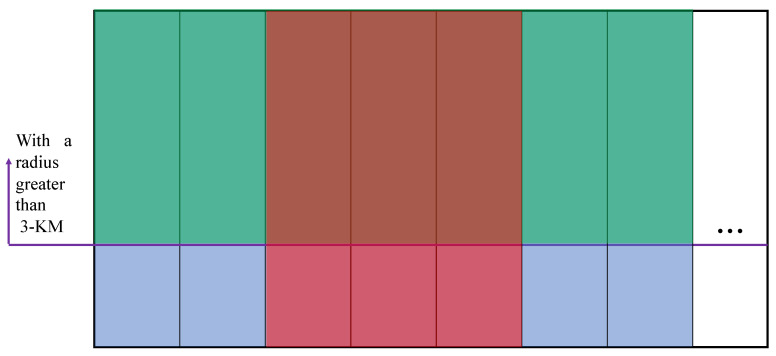
Visualization of the relationship between the existing crater datasets.

**Figure 15 sensors-21-00959-f015:**
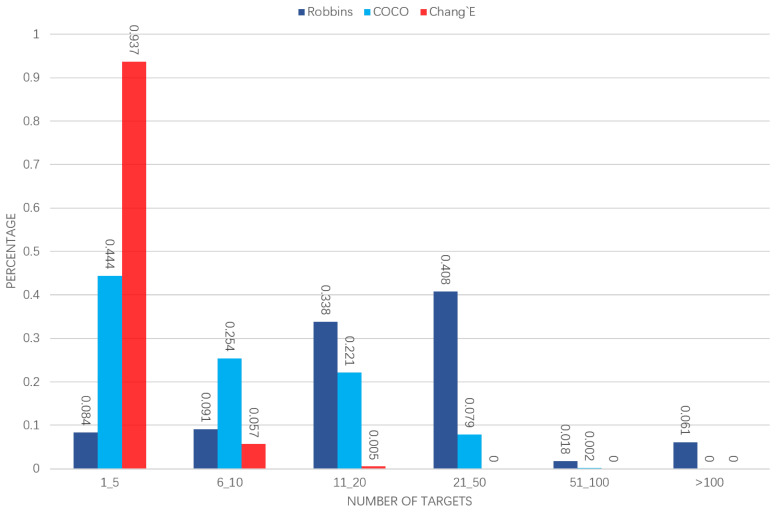
The distribution of the target number in each picture (512×512 pixels) on the three datasets. The x-axis represents several different ranges of the number of targets in each picture. The y-axis represents the proportion of the images containing the corresponding number of targets in the total dataset.

**Figure 16 sensors-21-00959-f016:**
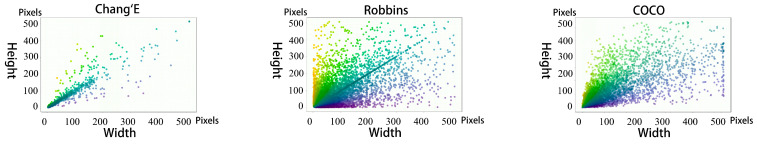
Comparison of the distributions of aspect ratio in Chang’e, Robbins and COCO datasets.

**Figure 17 sensors-21-00959-f017:**
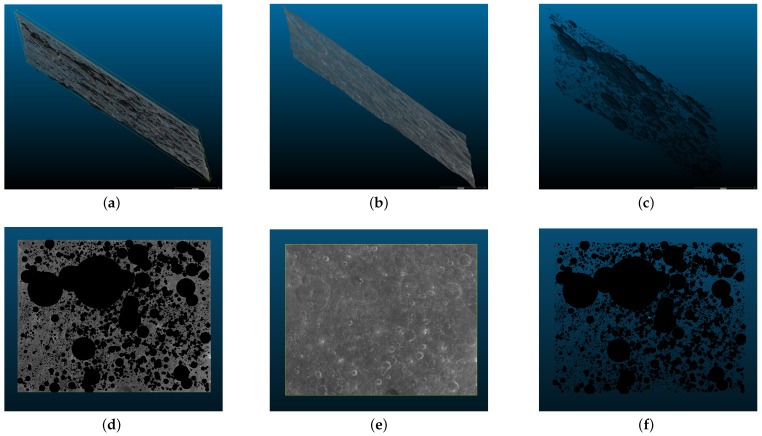
The three-dimensional structure of CE2_GRAS_DEM_50m_K003_49S120W_A, where (**a**) is all point clouds with crater semantic information, (**b**) is all point clouds without semantic information and (**c**) is only crater point clouds. (**d**–**i**) are the display of (**a**–**c**) from different perspectives.

**Figure 18 sensors-21-00959-f018:**
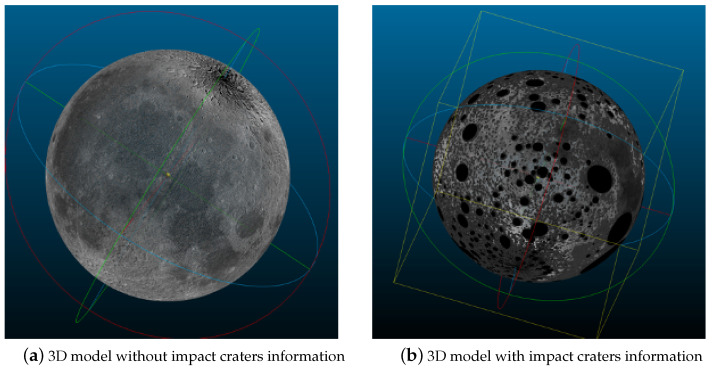
Visualization after 4096 times downsampling of all point clouds. (**a**) presents a 3D model without impact craters information, and (**b**) presents a 3D model with impact craters in black.

**Table 1 sensors-21-00959-t001:** Time taken for different annotation methods in Chang’e data.

Annotating Type	Time	Number of Pictures	Average Time
Complete	33 min	50	39 s
Auxiliary	17 min	50	20 s
Only observe	13 min	50	15 s

**Table 2 sensors-21-00959-t002:** Detection experiments on COCO and Red crater datasets using SSD networks. The Red crater dataset is our labeled dataset, excluding other labeled data.

Model	Data Shape	Batch Size	mAP of Crater	mAP of COCO
vgg16 atrous	300	32	0.5858	0.251
resnet50 v1	300	16	0.5583	0.306
vgg16 atrous	512	16	0.5538	0.289
mobilenet1	512	32	0.5426	0.217
resnet101 v2	512	4	0.5255	*

## Data Availability

Data available in a publicly accessible repository that does not issue DOIs. Publicly available datasets were analyzed in this study. This data can be found: https://drive.google.com/file/d/175f68A9L-wvvzYLc3X0DQd_SpxX5Geu0/view?usp=sharing.

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
