# Peer review of "Three-Dimensional Model of the Moon with Semantic Information of Craters Based on Chang’e Data"

_sensors, 2021, doi:10.3390/s21030959_

Round 1

Reviewer 1 Report

The paper analyses  the lunar mapping data provided by the Chang’E lunar exploration project. The authors present an algorithm for the detection of impact craters in the Chang’E dataset based on computer learning techniques. They discuss a thus obtained crater data set and compare it to other existing data sets. Finally, they discuss 3-D mapping techniques for their data set and illustrate them with 3-D visualizations.

The task of the reconstruction of the impact crater data set from high resolution mapping data such as provided by the Chang’E mission is timely and important for lunar science in general. The use of computer learning techniques for such a project as suggested by authors is adequate in light of recent development in this field, and the authors demonstrate that they had success with it. But the reviewer has found the language of the paper difficult to understand, and in some places he had to guess the intended meaning. The introduction lacks the discussion of other lunar crater reconstruction projects and of other lunar topography data sets. The authors could at least mention the NASA Lunar Reconnaissance Orbiter (LRO) mission. In the reviewer's opinion the presentation lacks focus. Though the sequence of tasks necessary to produce a crater data set is mentioned several times, the algorithm is never clearly described. It is not clear if a single method or several competitive methods were used. The description gives an impression that the detection of each crater required human interaction, but that seems to be implausible considering the amount of data that had to be processed. This needs to be clarified. The paper would benefit from a shorter, but more focused description of the crater detection algorithm with each step precisely defined with a reference to the appropriate software used/developed. No reference is provided for either software developed for this project or for the derived crater data set. So, it is not clear how the scientific community in general will benefit from this project. Overall, the reviewer believes that sufficient amount of work has been done as part of this project, and this research merits a publication, but only after the language and the structure of this paper are substantially improved. 

The following line by line comments contain some suggestions on improving the structure of the paper.

Line  1: If a statement is made that Chang’E mission provides the “highest resolution” data for DOM and DEM of the Moon, this definitely needs to be explained in the introduction. One should, at least, mention the resolution of the data set being used and compare it to the resolution of other existing data sets.

Lines 29-33: It is not clear why authors mention the Apollo mission in relation to the lunar mapping. The Apollo mission was about human landing and sample return. If one means the images that were taken in preparation for this mission, then that was the Lunar Orbiter Image Recovery Project (LOIRP). But that was more than 50 years ago. If one wants to make a comparison, one should certainly use modern missions, like  NASA Lunar Orbiter Laser Altimeter (LOLA).

Figure 1 : This figure doesn’t provide any useful information. If the goal was to demonstrate the high resolution of the DOM image, at least its scale should be shown. The Dem image is just binary, it doesn’t provide any information on its vertical distribution, and without the scale there is also no information of the horizontal resolution.

Line 50 : The authors use an expression “semantics of the impact crater“ in many places in the paper, but they never explain what they mean by that.

Lines 60-76 : It is not clear why file naming conventions need to be described in the main body of the paper. Seems like moving them to the appendix would be a better choice (this is just a suggestion). 

Lines 78-82: The language of this paragraph needs to be improved. What is the meaning of the sentence “The binary stream of the edited image is characterized in the JSON data.”? One is not editing the image, but just extracts information from it.

Lines 84-85 : It is not clear why one would need to apply a crater detection algorithm twice.

Lines 128-140 : The Chang’E-4’s rover data are irrelevant to the research presenter in  this  paper,  so it is not clear why they are mentioned here.

Figure 6 : The meaning of the presented sequence is not clear. If these are “modules” of the existing framework as indicated on the line 142, then they can’t include “implementation”. If this is a sequence of tasks, then why would one need to detect craters in box 3 if they were already marked in box 2?. What is meant by “crater reconstruction” in box 4? If one is dealing with a DEM, then once the location of a crater has been detected, there is nothing else to reconstruct. Are authors talking about the visualization here?

Lines 170-175 : This section is not clear. The use of mathematical notations is confusing. One should either adopt more formal mathematical language or not use it at all. 

Line 209 : Is an extra “x G” missing in the formula?

Figure 11 : Should the right-bottom box have H2 x W2 x C2 ? 

Lines 205 - 216 : Grouping seems to be an important part of the algorithm, but no description (or reference) is provided on how the grouping is done.

Lines 286-288 : It is not clear what “observe set” option is. Is it about the operator loading the data set and not doing anything with it? Why would it take any time?

Line 296 : One needs to provide a reference for the external data set used. Just calling it a “NASA” data set doesn’t identify it properly.

Figure 18 : The purpose and meaning of the figure 18 is not clear. What is represented by the horizontal sequence of boxes?

Figure 19 : This figure is not explained properly. What are the units used on this figure? What is the “interval” the number of target frames are normalized to?

Figure 20 : This figure seems to provide useful information on the obtained data set. But no explanation of the results is given. Why are Chang'E craters smaller and have aspect ratio closer to one than two other data sets? What are the units on “height” and “width”?

Line 330 : What is meant by the expression “fluctuation of the impact crater” ?

Lines 334-337 : Statements (1)-(3) should be expressed in a more clear language. Meaning of  (2) is completely obscure.

Author Response

Dear Reviewer:

Thank you for your letter and comments concerning our manuscript entitled “Three-dimensional model of the moon with semantic information of

craters based on Chang’E’s data” (Manuscript ID: sensors-1015649). Those comments are all valuable and very helpful for revising and improving our paper, as well as the important guiding significance to our researches. We have studied comments carefully and have made correction which we hope meet with approval. Revised portion are marked in red in the paper. The main corrections in the paper and the responds to your comments are in the coverletter.

Reviewer 2 Report

A review of ‘Three-dimensional model of the moon with semantic information of craters based on Chang’E’s data’ by Yunfan LU et al.

The paper lacks at this stage sufficient scientific rigor to be acceptable as a paper in Sensors. The main reason is that quite some essential information is lacking. The authors describe what they call an ‘algorithmic process’. But an algorithm (process or not) is an implementation of a scientific method. This method should be clear and well described. At present that is not the case. For instance something like manual marking is unclear. Also, why a time T is associated to this, and what the Hcompletely and Tproposal have to do with this is unclear. Hence, in all it remains obscure how exactly the deep learning methodology is applied. That means: what are the test sites, and with which set-up was the analysis done. Particular choices on parameter values have been made, but they are not provided in the text and why were they made the way it was done? The description in section 3.1.3 is too short to be fully informative.

  • No list of references was provided in the end: hence all references are incomplete in the text.
  • Section 3.1.4. ‘Differences between Convolution and grouped convolution’ is strange. The authors clarify here why they make a particular choice. However the differences might be discussed in the discussion section, not in the methodology section. In that section, the chosen way of analysis should be properly justified and described.
  • Possibly, a sharp distinction in the manuscript between the methodology on the one hand and its implementation for the specific study could be helpful to make the manuscript better understandable and more complete.

Minor issues

  • The user interface with the LabelMe software (Figure 3) should be removed.
  • Hard- and software issues should be moved to the appendix. In particular, the way of labelling and algorithms are redundant.
  • The English is by times hard to follow, and contains a range of grammatical flaws. The manuscript should be inspected by an English native speaking scientist.
  • The acronym COCO is not explained, neither is mAP.

To my opinion, the manuscript has some merits: the story of the pipelines, the use of deep learning, the use of quadtrees and the general objective are interesting scientific angles for this study. At the moment, however, the set-up of the paper is too sloppy and has too many gaps to convince. Also, redundant information is to be removed and a much more transparent manuscript should emerge.

Reviewer 3 Report

The Chang'E2 DTM and ORI (called DOM here) are a goldmine of new information on the shape and morphology of the lunar surface. However, the DTMs and ORIs are not in a consistent global reference system such as that provided by the SELENE and LOLA (Barker, M. K.; Mazarico, E.; Neumann, G. A.; Zuber, M. T.; Haruyama, J.; Smith, D. E. (2016) A new lunar digital elevation model from the Lunar Orbiter Laser Altimeter and SELENE Terrain Camera. Icarus, 273, 346–355.). The claim that the global datasets of craters are inaccurate is, therefore, an assertion and remains unsubstantiated unless the authors can provide convincing evidence.

The Chang'E2 datasets are publicly available but there is no discussion of making the derived craters publicly available. The authors should be encouraged that they should make them available for publication and peer review as otherwise, they will not be useful to the world's lunar science community.

The most serious deficiency is the lack of any attempt to assess the accuracy and veracity of the resultant dataset. Without such measures such as the F1 score, it is extremely unlikely that any lunar scientist would use such a dataset.

The quality of Figure 1 is extremely poor. The authors should be encouraged to download QGIS and use the colorized-by-height hill-shading for the DTM as well as ensure that the DTM and DOM show the same area.

There is no discussion on the time taken for labeling the crater dataset and its reliability such as discussed by Francis, A.; Brown, J.; Cameron, T.; Clarke, R. C.; Dodd, R.; Hurdle, J.; Neave, M.; Nowakowska, J.; Patel, V.; Puttock, A.; Redmond, O.; Ruban, A.; Ruban, D.; Savage, M.; Vermeer, W.; Whelan, A.; Sidiropoulos, P.; Muller, J.-P. (2020) A Multi-Annotator Survey of Sub-km Craters on Mars. Data, 5, 70.

Drawing an analogy between a rover traverse GPR (Ground Penetrating Radar) and lunar-wide subsurface mapping seems like rather a stretch. A reference should be added to the subsurface mapping studies from Kaguya/SELENE of Ono, T.; Kumamoto, A.; Nakagawa, H.; Yamaguchi, Y.; Oshigami, S.; Yamaji, A.; Kobayashi, T.; Kasahara, Y.; Oya, H. Lunar Radar Sounder Observations of Subsurface Layers Under the Nearside Maria of the Moon. Science 2009, 323, 909.

Line 168, what is the multiplier of?

lines 181-182, what does an experimental principle refer to?

In Figure 20, what does the aspect ratio refer to? The usual measure is depth-to-diameter and why should one expect them to be linearly correlated?

See highlighted areas below

sensors-1015649-peer-review-v1 - annotated

produced

Page 1

Dec 6, 18:08, by Anonymous

And impact craters are the most basic feature of the moons surface.

Page 1

Dec 6, 18:08, by Anonymous

Will this be made publicly available?

Page 1

Dec 6, 20:09, by Anonymous

Introduction

Mastering these laws, geologists can not only understand the structure and motion properties of topography and landforms at different scales but also predict and analyze future changes.

Page 1

Dec 6, 20:09, by Anonymous

This DEM figure is far too poor quality. Please show a hill-shaded DTM (use QGIS) of the equivalent area alongside mapping information on the location.

Page 2

Dec 6, 20:09, by Anonymous

Is this tool publicly available like the ChangE’2 DOM+DEM products? If so, where can it be downloaded from?

Page 2

Dec 6, 20:09, by Anonymous

Is this co-registered to the global laser altimeter dataset (SELENE+LOLA)?

Page 2

Dec 6, 20:09, by Anonymous

In the 3D model reconstruction, we use a standard sphere as an approximation to restore the height of each point, to build a more intuitive and beautiful moon 3D model.

Page 2

Dec 6, 20:09, by Anonymous

Related Work - Lunar impact crater detection algorithm - Impact detection based on digital methods

Existing methods use a wide range of high-precision data to detect craters using spatial analyzing and digital image processing (e.g., shape tting-based algorithm, Hough transform-based algorithm

Page 4

Dec 6, 20:09, by Anonymous

Related Work - 3D Construction of the moon model

The work used data obtained from Lunar Penetrating Radar of ChangE-4 to recover the geological information of ChangE-4 walking under the road.

Page 5

Dec 6, 20:09, by Anonymous

Methods - Auxiliary Marking Method and Analysis.

151 152 153 154 155 156 157 158 159 160 161 162 163 164 165 166 167 168 169 170 171 172 173 174 175 However, since ChangE-4 is a ground robot, the scale of the obtained data is very small (only a small area near the moon 45.4446 S, 177.5991 E). Instead, We use the full-coverage-data scanned by ChangE-1 and ChangE-2 which have a comprehensive and high-level understanding of the moon topography. 3. Methods Our framework is composed of four modules and nally reconstruct impact craters. We give our pipeline in Fig. 6. During the whole workow, we achieve the implementation of the marking tool, marking of the impact crater, impact crater detection, and impact crater reconstruction in turn. Figure 6. The pipeline of our Lunar geological reconstruction framework. 3.1. Auxiliary Marking Method and Analysis. The full markup tools represented by LabelMe [17] have been widely used in object detection, semantic segmentation, instance segmentation, and other visual understanding tasks of data marking. The full mark method is a method of marking all object areas in an image from scratch. The process is simply entering a picture and marking it with tools like LabelMe. However, the full markup method is very redundant for large-scale data. Thus, we use the auxiliary marker method [1] to accelerate subsequent large-scale crater markings. We implement a neural network-based assisted labeling method.

Page 6

Dec 6, 20:09, by Anonymous

Are there no plans to have an orbiting spacecraft to map the lunar subsurface down to 40m?

Page 6

Dec 6, 20:09, by Anonymous

We give our pipeline in Fig.

Page 6

Dec 6, 20:09, by Anonymous

Increased the average image marking time from 14s to 5s.

Page 6

Dec 6, 20:09, by Anonymous

Multiplier of what?

Page 6

Dec 6, 20:09, by Anonymous

Methods - Auxiliary Marking Method and Analysis. - The Algorithm Processes

While for the auxiliary process, The time can be express as T proposal= Time( P( I ), H proposal).

Page 6

Dec 6, 20:09, by Anonymous

While for the auxiliary process, The time can be express as T proposal= Time( P( I ), H proposal).

Page 6

Dec 6, 20:09, by Anonymous

In the case of consistent accuracy, the sum of the time of P(i ) and adjusting should less than the time of completely marking.

Page 6

Dec 6, 20:09, by Anonymous

Methods - Auxiliary Marking Method and Analysis. - Quantification

What does this mean?

Page 7

Dec 6, 20:09, by Anonymous

Specically, it can be divided into serveral E, F processes;

Page 7

Dec 6, 20:09, by Anonymous

It can be seen from the actual operation the there is no method for parallelizing pipeline stages in the traditional completely method.

Page 7

Dec 6, 20:09, by Anonymous

It can be seen from the actual operation the there is no method for parallelizing pipeline stages in the traditional completely method.

Page 7

Dec 6, 20:09, by Anonymous

Experiments - Dataset Analysis

Reference required to be added.

Page 13

Dec 6, 20:09, by Anonymous

The published data set can be obtained through NASA, so we call it NASA, most of the impact craters in this part of the data are more than 3KM.

Page 13

Dec 6, 20:09, by Anonymous

What do the columns represent?

Page 13

Dec 6, 20:09, by Anonymous

What does this aspect ratio refer to as craters are all circular? What do the different colours refer to? The plot needs a colour key.

Page 14

Dec 6, 20:09, by Anonymous

Experiments - Create 3D model

If the point belongs to For a certain impact crater, the point is marked to distinguish impact craters from non-impact craters.

Page 16

Dec 6, 20:09, by Anonymous

Author Response

(The authors gave the same response as above.)

Round 2

Reviewer 2 Report

The manuscript is much improved following the earlier version. The readability should improve by a better use of the scientific language, in particular for the newly added sections. For instance, section 4.1 is very hard to follow: sometimes plural should be used in stead of singular phrasing and there is an abundant use of brackets - which should be avoided. If carefully dealt with, the interest of the readership will increase.

Reviewer 3 Report

It appears I cannot upload separate files so here are my comments on their cover latter. I can send you these separately by email.

This should be a condition of publication that at a bare minimum all authors should publish their tool and their datasets. Most high impact journals (e.g. ESSD, ESS) insist that datasets and software should be made available at the time of publication. At the very least the tool and a subset of the data should be released as supplementary materials.

Page 1

Jan 4, 14:24, by Anonymous

The quality of the orthoimage, in particular, is too poor for publication. I recommend that the authors follow the guidance provided. All images should be contrast stretched.

Page 1

Jan 4, 14:24, by Anonymous

This is not science to state that because someone was trained it is good enough. An evaluation needs to be made and the results presented as proposed.

Page 1

Jan 4, 14:25, by Anonymous

The Chang e’2 is not co-registered to any global reference frame. This means that you will not be able to compare one set of crater locations with another if these other datasets use the SELENE+LOLA as a basemap. You should state that you do everything using Chang e’2data and not in a global reference system.

Page 3

Jan 2, 10:48, by Anonymous

sensors-1015649-peer-review-v2 - annotated copy

Abstract: Chinas ChangE lunar exploration project obtains the Digital Orthophoto Image data (DOM) and the digital elevation model (DEM) data covering the whole moon, which have great signicance for lunar researches.

Page 1

Jan 1, 18:25, by Anonymous

has produced

Page 1

Jan 1, 18:25, by Anonymous

a

Page 1

Jan 1, 18:25, by Anonymous

Introduction

Contrast needs to be enhanced of the DOM image.

Page 2

Jan 1, 18:25, by Anonymous

Restoring the three-dimensional structure of the moons surface and showing the specic location of the impact crater can get more intuitive results.

Page 2

Jan 1, 18:25, by Anonymous

each

Page 2

Jan 1, 18:25, by Anonymous

Related Work - Robust Annotation Tool.

What does this word mean?

Page 3

Jan 1, 18:25, by Anonymous

Provide several references to papers which I use this tool to justify your statement.

Page 3

Jan 1, 18:25, by Anonymous

At the same time, is also has a variety of annotation methods.

Page 3

Jan 1, 18:25, by Anonymous

We can use LableMe to annotate images with polygon, rectangle, circle, line and point.

Page 3

Jan 1, 18:25, by Anonymous

Methods

3.2. Third, we used a quadtree decomposition method based on the deep learning detection methods to detect impact crater (more details in Section

Page 4

Jan 1, 18:25, by Anonymous

3.3). Finally, we created a 3D model with impact craters information.

Page 4

Jan 1, 18:25, by Anonymous

Methods - Construction of the 3D model with impact craters information based on GIS coordinate mapping

taken

Page 11

Jan 1, 18:25, by Anonymous

Cumber of pictures

Page 11

Jan 1, 18:25, by Anonymous

Experiments - The Relationship between mAP and Auxiliary Cost

During the annotating stage, we average the sum time of all three annotators when each annotator annotates three sets of photos which are completely set (label an image without any supplementary information), auxiliary set (input is the result of a proposa

Page 11

Jan 1, 18:25, by Anonymous

on

Page 11

Jan 1, 18:25, by Anonymous

Tab.

Page 11

Jan 1, 18:25, by Anonymous

Table

Page 11

Jan 1, 18:25, by Anonymous

Experiments - Crater detection

of diameters?

Page 13

Jan 1, 18:25, by Anonymous
